# A “Population Dynamics” Perspective on the Delayed Life-History Effects of Environmental Contaminations: An Illustration with a Preliminary Study of Cadmium Transgenerational Effects over Three Generations in the Crustacean *Gammarus*

**DOI:** 10.3390/ijms21134704

**Published:** 2020-07-01

**Authors:** Pauline Cribiu, Alain Devaux, Laura Garnero, Khédidja Abbaci, Thérèse Bastide, Nicolas Delorme, Hervé Quéau, Davide Degli Esposti, Jean-Luc Ravanat, Olivier Geffard, Sylvie Bony, Arnaud Chaumot

**Affiliations:** 1INRAE, UR RiverLy, Laboratoire d’écotoxicologie, F-69625 Villeurbanne, France; pauline.cribiu@developpement-durable.gouv.fr (P.C.); laura.garnero@inrae.fr (L.G.); khedidja.abbaci@inrae.fr (K.A.); nicolas.delorme@inrae.fr (N.D.); herve.queau@inrae.fr (H.Q.); davide.degli-esposti@inrae.fr (D.D.E.); olivier.geffard@inrae.fr (O.G.); 2ENTPE, INRAE, CNRS UMR 5023 LEHNA, rue Maurice Audin, 69518 Vaulx-en-Velin CEDEX, France; alain.devaux@entpe.fr (A.D.); therese.bastide@entpe.fr (T.B.); sylvie.bony@entpe.fr (S.B.); 3CEA, LAN, 17 rue des Martyrs, 38054 Grenoble CEDEX 9, France; jean-luc.ravanat@cea.fr

**Keywords:** Crustaceans, Amphipoda, *Gammarus*, cadmium, transgenerational effects, parental exposure, population effects, life-history trait, demographic impacts, population dynamics modeling

## Abstract

We explore the delayed consequences of parental exposure to environmentally relevant cadmium concentrations on the life-history traits throughout generations of the freshwater crustacean *Gammarus fossarum*. We report the preliminary results obtained during a challenging one-year laboratory experiment in this environmental species and propose the use of population modeling to interpret the changes in offspring life-history traits regarding their potential demographic impacts. The main outcome of this first long-term transgenerational assay is that the exposure of spawners during a single gametogenesis cycle (3 weeks) could result in severe cascading effects on the life-history traits along three unexposed offspring generations (one year). Indeed, we observed a decrease in F1 reproductive success, an early onset of F2 offspring puberty with reduced investment in egg yolk reserves, and finally a decrease in the growth rate of F3 juveniles. However, the analysis of these major transgenerational effects by means of a Lefkovitch matrix population model revealed only weak demographic impacts. Population compensatory processes mitigating the demographic consequences of parental exposure seem to drive the modification of life-history traits in offspring generations. This exploratory study sheds light on the role of population mechanisms involved in the demographic regulation of the delayed effects of environmental toxicity in wild populations.

## 1. Introduction

Accumulating evidences from wildlife populations and experimental studies demonstrate that environmental stressors adversely influence populations beyond exposed generations. The time lag between environmental exposure and biological impacts challenges current ecological risk assessment [1], since delayed effects may deeply affect the future ability of populations to persist in their environment [2,3]. Ecotoxicological studies have now highlighted that delayed effects of contaminants are involved in the persistence or aggravation of the long-term adverse impacts of chronic contaminations [4,5], but also in the development of tolerance [6,7]. The long-term effects of chemical contaminations also influence the population vulnerability to cope with additional environmental stressors [7,8]. It is noteworthy that the majority of these ecotoxicological studies dealing with the role of delayed effects in long-term impacts of contamination have focused on trace metals. On the other hand, some authors have proposed that the delayed effects on life-history traits occurring after environmental changes (e.g., food privation, chemical exposure, etc.) may regulate population demography and facilitate population sustainability [9]. The demographic impact of trait alterations may be mitigated by life-history tradeoffs during the lifetime of individuals or across generations [10,11]. At the population level, such tradeoffs may sustain demographic compensation processes [12]. Species evolutionary history potentially constrains these tradeoffs as illustrated by environmental canalization phenomena [13]. In this context, the use of population dynamics models has already shown a substantial interest to assess the demographic consequences of life-history trait impairments in multigenerational exposure designs [11,14,15].

Life-history trait alterations after parental exposure to chemicals are frequently documented in F1 generations, notably in arthropods [16,17,18,19], and the proposals for considering F1 generations in standard toxicity protocols are increasing [4,20]. Nevertheless, few long-term studies in aquatic ecotoxicology have specifically demonstrated transgenerational effects, i.e., effects in generations that were never exposed at any stage of their life, including germ cell and embryo stages. Most of them were carried out on laboratory species such as *Danio* [21], *Daphnia* [22,23], or *Caenorhabditis* [24]. More broadly, multigenerational experiments are mostly restricted to laboratory model species selected for the shortness of their lifetime. As a result, the consequences of parental exposure along offspring generations are still poorly explored in the variety of species from aquatic animal biodiversity, the majority of which have longer lifetimes than laboratory model species. It is very likely that peculiar patterns related to specific tradeoffs in energy allocation and diverse demographic strategies occur in the long-term responses of these populations to contamination.

Crustacean gammarids are a major constituent of animal communities inhabiting European freshwater lotic ecosystems. They are increasingly used as bioindicators in environmental monitoring [25]. Metallic and especially cadmium (Cd) contaminations have been documented as a limiting factor of gammarid abundance in French watercourses [26,27]. Besides, we recently showed in one field *Gammarus fossarum* population historically exposed to Cd that transgenerational processes were involved in the modification of toxicological sensitivities and their related fitness costs [7,28]. In addition, the exposure to low Cd concentrations in the laboratory modulated epigenetic marks in *G. fossarum* [29]. Cd transgenerational effects were consistently reported in several other invertebrates [30,31,32]. In addition, delayed effects of parental genotoxic stress mediated by DNA damage in spermatozoa have been demonstrated in *G. fossarum* F1 offspring [33]. This study aims to further explore the population effects of *G. fossarum* Cd exposure during the reproductive cycle by investigating for the first time its impact on future generations. We investigated the consequences of a short parental exposure to environmentally relevant Cd concentrations in terms of genome integrity, life-history traits, and demography. To this end, we attempted to carry out a challenging one-year laboratory experiment to monitor the delayed effects up to the third generation of offspring reared under uncontaminated conditions after F0 exposure. Then, we analyzed the life-history traits of F1, F2, and F3 generations with a Lefkovitch matrix model [34] in order to assess potential demographic consequences. Thus, by getting an integrated vision at the population scale of the multigenerational response to initial exposure, we formulate here the new hypothesis that population regulatory mechanisms may shape the transgenerational effects of contaminants on life-history traits in this freshwater sentinel species.

## 2. Results and Discussion

### 2.1. Experimental Design of the Transgenerational Study

The overall experimental design established according to our previous knowledge of the reproductive cycle [35,36] and population dynamics of *G. fossarum* [34] is summarized in Figure 1.

F0 adult gammarids were exposed during one reproductive cycle to two environmentally relevant concentrations of Cd (0.3 and 3 µg Cd/L). These concentrations were fixed in the range of concentrations used in previous studies [35,37,38,39] seeking to induce only sublethal effects (reproductive disorders) in F0 spawners. Thereafter, control, 0.3 µg Cd/L, and 3 µg Cd/L parental exposure conditions are indicated by using the abbreviations C-F0, 0.3Cd-F0, and 3Cd-F0, respectively. To apply the exposure during a single gamete maturation cycle, the exposed parents were selected in precopulatory pairs with females in the D2 pre-molt stage (i.e., just before the ecdysis of females, which marks the start of a new reproductive cycle), and the exposure was stopped before the laying and the fertilization of oocytes in the marsupium (Figure 1). Then, F0 parents with F1 embryos developing in marsupia were maintained in uncontaminated water until they released neonate juveniles. F1 organisms were bred in Cd-free conditions. F2 and F3 generations were bred in the same conditions. The experiment was stopped after 58 weeks, just before F3 offspring reached puberty due to technical problems in our water supply facilities.

We started the multigenerational test from a large number of F0 breeders (400 couples per condition) in order to be able to breed three generations over more than a year (Figure 1). We favored exposure in large volume containers (12 L buckets) to manage such a large number of F0 spawners at the expense of setting up exposure replicates; only replicates of biological measurements were implemented (Appendix A). We opted for this dose–response design with pseudo-replication at the individual level only, because we had a very good experience of the exposure protocols used, as well as previous knowledge of the expected effects of these cadmium levels in terms of feeding inhibition, disruption of the molting cycle, vitellogenesis, spermatogenesis, and sperm genotoxicity in this species [35,37,38,39]. However, the production of F1 neonates was lower than expected. This forced us to work with small numbers of organisms when rearing offspring generations (Appendix A). As a first consequence, we had to compensate for imbalances in densities between rearing tanks by regularly redistributing organisms to maintain comparable densities between tanks and avoid bias due to density effects on life history traits. As a second consequence, the numbers of F2 neonates produced in the condition 0.3Cd-F0 were not sufficient to pursue the experiment for this condition. We are aware that the final design with individuals as the only pseudo-replicates of exposure limits the significance of the results obtained to conclude on Cd population effects. Nevertheless, these first observations of long-term effects of Cd exposure on the scale of several successive generations in an environmental species are unprecedented. Therefore, they constitute an initial set of valuable data to describe first response patterns and on which new hypotheses for understanding population impacts can be based.

### 2.2. Toxic Stress of Cd Exposure in F0 Spawners

As expected, no effect was observed on F0 survival whatever the Cd concentration (prop.test, *p*-value = 0.53). Regarding reproduction endpoints, the females exposed to Cd exhibited a disruption of their progress in the reproductive cycle with 40% of females at AB-C1 molt stages compared to control at the end of exposure (Figure 2A). That confirmed the molting delay reported by Geffard et al. (2010) in *G. fossarum* exposed to similar Cd exposure in the laboratory. The mean oocyte number of females exposed to the highest Cd concentration was lower than that of controls (Figure 2B). Nevertheless, the difference was not significant (Kruskal–Wallis test (KW), *p*-value = 0.089) with high inter-individual variability under contamination conditions. An absence of gonadic development was observed in 10% of exposed F0 females to 0.3 µg Cd/L and in 40% of those exposed to 3 µg Cd/L. Such levels of molting delay and fecundity reduction are in line with those recorded in *G. fossarum* during previous in situ caging studies conducted in metallic field contamination contexts [40]. No significant difference was observed in feeding rate and mean secondary follicle surfaces per female (KW, *p*-value = 0.14 and *p*-value = 0.90, respectively) (Appendix A). Hence, the dietary intake and the energy allocation to eggs in females that achieved the onset of vitellogenesis were likely not affected by the exposure. This discards the hypothesis of feeding limitation that could have been advanced to explain the molting delay and the reduction of oocyte number as already reported in this species [41]. Instead, the high percentage of F0 females without oocytes could be explained by a toxic effect of Cd on gametogenesis such as a disruption of follicle recruitment or a lethal effect on the provisioned eggs. Primary DNA damages were significantly higher in the sperm of exposed F0 males independently of the Cd concentration compared to control (KW, *p*-value = 9.2 × 10^–4^) with mean tail intensities ranging between 10% and 20% (Figure 2C). No effect on sperm viability was observed (KW, *p*-value = 0.46) (Appendix A). These levels of primary DNA damages in F0 sperm cells reach the upper range of genotoxicity levels observed in field monitoring studies using caged male *G. fossarum* as a probe of genotoxicity in different French watercourses, including metallic contamination contexts [42]. Global cytosine methylation levels were not different between controls and gammarids exposed for 21 days to Cd (KW, *p*-value = 0.40) (Appendix A). Notably, our previous results showed a fluctuating trend of this marker at similar Cd concentrations, with a hypomethylation after 14 days of exposure followed by a weak hypermethylation after one month [29].

Overall, the exposure did cause mainly reproductive disorders of F0 male and female gametogenesis. These impacts corresponded to similar levels as toxic effects recorded in field contaminated aquatic environments. These impairments were unlikely related to a disruption of energy intake. All of these initial results on reproductive disruption in F0 spawners supported the appropriateness of the exposure protocol chosen to investigate the delayed population effects that could occur on future generations.

### 2.3. Cascading Effects on F1, F2 Reproductive Features and F3 Growth Ability

The survival of F1 juveniles from 3Cd-F0 was similar to controls (prop.test, *p*-value = 0.99). By contrast, the survival of F1 juveniles from 0.3Cd-F0 was significantly lower than in control (prop.test, *p*-value = 9.5 × 10^–3^; Appendix A). However, this 11% decrease was weak in regard to the variability of juvenile survival in natural populations [34]. In addition, the experimental setup was based on only one experiment without biological replicates, which further relativizes the variability in the survival rate in relation to the exposure condition. No difference in F1 growth rates was detected between control and F0-exposed conditions (KW, *p*-value = 0.44) (Appendix A). In line with the observation that the investment in eggs was not altered in F0 females, these results reinforce the study of Lacaze et al. [33], who showed in *G. fossarum* that primary DNA damages below a threshold of 20% tail DNA intensity in spermatozoa did not lead to impairments in the embryonic development. This finding contrasts with the delayed effects observed in other species—for instance, on the hatching success in the F1 generation of the freshwater gastropod *Physa pomilia* after F0 exposure to 2.5 µg Cd/L and 10 µg Cd/L [43]. Regarding the adult stage, a significant decrease in brood size of F1 females from 3Cd-F0 was observed (*p*-value = 1.8 × 10^–3^) (Figure 3A). Moreover, we observed F1 females without any embryo in their marsupia in Cd-exposed conditions. However, parental exposure had no significant effect on oocyte production, nor on secondary follicle surface in F1 females (KW, *p*-value = 0.34 and *p*-value = 0.59, respectively) (Appendix A). Hence, the reduction observed in brood size might correspond to a disruption of laying, fertilization, or early embryo development phases. Contrasting with the present study, the parental exposure to Cd concentrations of 1, 5, and 10 µg/L during gametogenesis did not affect the reproductive capacity of F1 offspring in Japanese medaka *Oryzias latipes* [44]. In the same way, Guan and Wang [22] showed that the reproduction of F2 adult *Daphnia magna*, corresponding to F1 in the present study, was not impacted by parental Cd exposure. Parental exposure had no significant impact on the feeding rate of F1 males (KW, *p*-value = 0.43) (Appendix A), nor on sperm viability and primary DNA damages in sperm cells (KW, *p*-value = 0.50 and *p*-value = 0.088, respectively) (Appendix A). No difference in global cytosine methylation levels was observed between F1 from control and F1 from 0.3Cd-F0 and 3Cd-F0 conditions, but a significant increase was detected in F1 from 3Cd-F0 compared to F1 from 0.3Cd-F0 (KW, *p*-value = 0.021) (Appendix A).

Overall, the main effect of parental Cd exposure on F1 was a decrease in embryo production. Different mechanistic hypotheses could explain such a reproductive impairment of the generation F1. First of all, our exposure design does not exclude the possibility of a direct deleterious impact of Cd on the early development of F1 organisms due to a potential Cd transfer from F0 mothers to their eggs [32,45]. In view of the absence of growth effect, the hypothesis of an endocrine disruption during development with delayed reproductive effect at the adult stage could be privileged here against that of a disruption of energy allocation in the F1 organisms. The possibility of an endocrine disruptive effect of Cd has already been advanced in crustaceans, notably in freshwater crustaceans [46]. This hypothesis is based either on some physiological observations of disturbance of molting and reproduction in *G. fossarum* for instance (inhibition of secondary vitellogenesis and molting delay) [35], or more specifically via different observations of a possible disturbance of hormonal signaling, for example of the ecdysteroid pathway in crabs acutely exposed to Cd [46]. Jaegers and Gismondi [47] also suggested recently that a low concentration of Cd (1 µg/L) could impair the endocrine system of *Gammarus pulex* organisms via alteration of the methylfarnesoate signaling pathway. Methyl farnesoate is a hormone implied in molting, the control of oogenesis, and possibly in testicular development in crustaceans. In support of this hypothesis, we also showed in *G. fossarum* using a targeted proteomics approach that chronic exposure to 2 µg/L of Cd could inhibit the production of an enzyme specifically involved in the metabolism of this hormone [39]. However, all of these results, which suggest the possibility of an endocrine disruption action of Cd in the amphipods, have only been obtained in sexually mature individuals. Data of embryonic phase exposure are scarce in this species [48], and none of them have examined the delayed consequences of embryo exposures on reproduction in adult stages. The reproductive alteration in F1 might also result from the delayed consequence of altered genome integrity in F0 germ cells, as recorded in sperm cells by the comet assay (Figure 2C). Indeed, the absence of a significant reduction in F0 sperm viability under Cd exposure might lead to the transfer of possible impacts in offspring of the genotoxic effects observed in F0. The fertilization success of F0 was also not affected, as already observed in other invertebrates with similar Cd concentrations [49,50] or in fish [51,52]. In addition, a lack of DNA machinery repair in crustacean sperm cells has been suggested in different studies [37,53] and genotoxic effects of Cd on DNA machinery repair have been recorded [54,55,56,57].

No difference was recorded in F2 survival at the 8^th^ week after neonate collection, nor in F2 growth rates between conditions (prop.test, *p*-value = 0.17 and MW, *p*-value = 0.31, respectively) (Appendix A). In F2 organisms of the condition 3Cd-F0, the median time to reach puberty was strongly shortened as much as 3 weeks earlier than in organisms from the control condition (Figure 3B). The lag between C-F0 and 3Cd-F0 was 7 weeks when all the females reached maturity in each condition. Consistently, the body size at puberty was significantly reduced in 3Cd-F0 (MW, *p*-value = 1.7 × 10^–4^) (Figure 3C). Such accelerated pubertal development was previously reported in female rats in F3 generations (corresponding to F2 offspring in our study) stemming from an F0 exposed to atrazine [57] or to dioxins [58]. Rattan et al. [59] also showed that Di(2-ethylhexyl)-phthalate (DEHP) exposure of F0 accelerated the onset of puberty along three generations in mice, thereby including the generation that corresponds to F2 offspring in the present study. By contrast, few studies have explored such shifts of maturity in invertebrates. A rare example is the delay in the emergence of males in F0 generation and non-exposed F1 generation recorded in the marine mollusk *Crepidula onyx* after a 2,2′,4,4′-tetrabromodiphenyl (BDE-47) parental exposure [60]. As discussed above, endocrine disruption might be involved in the pattern response of reproductive alteration in F1, and it could be hypothesized that the early puberty observed in F2 *G. fossarum* is a consequence of this disruption. For instance, Leblanc et al. [9] showed the possibility of such transgenerational transmission of hormonal changes in the crustacean *Daphnia pulex*, which caused modifications in sex ratio and brood size in subsequent generations of exposed females to the endocrine disruptor insecticide pyriproxyfen. However, this mechanistic interpretation deserves now more extensive molecular studies.

Changes in the size of secondary follicles were also observed in the F2 generation with an average surface area of secondary follicles greater than 0.11 mm^2^ in two-thirds of F2 females of the condition 3Cd-F0, which is a value that is never reached in F2 females of the condition C-F0. However, the difference between the two conditions was not significant, which was probably due to the low sample size (MW, *p*-value = 0.10) (Figure 3D). No significant effects in the oocyte and embryo number of F2 females were recorded (MW, *p*-value = 0.86 and *p*-value = 0.53, respectively) (Appendix A). Nevertheless, as for the F1 generation, F2 females with no embryos were only observed in the condition 3Cd-F0 (29%). In addition, F2 males exhibited primary DNA damages significantly higher than those recorded in the control condition (MW, *p*-value = 2.3 × 10^–3^). However, they remained on a very low level (< 5%) considering the reference level of 3.5% defined in this species [42] (Appendix A). Overall, the Cd exposure of F0 parents resulted in an earlier puberty and an increase in egg abortion rates in F2 spawners. However, the average F2 fertility, i.e., number of embryos, was restored compared to the F1 generation.

In contrast with the generations F1 and F2, the F3 juveniles from Cd-exposed F0 showed a sharp lower growth rate than control (MW, *p*-value = 0.021) (Figure 4A), along with no significant difference in survival (survival rate of 100% until the 8^th^ week after F3 neonate collection). The body size of F3 organisms at the puberty onset (8^th^ week after F3 neonate collection) was affected by their great-grandparent Cd exposure (MW, *p*-value = 1.9 × 10^–3^) (Figure 4B). Hence, the modifications of energy allocation observed in F2 organisms (i.e., earlier puberty and reduction of egg reserves) translated in a lower growth ability of F3 offspring. Such cross generation trade-offs were previously reported in *Gammarus* [61], in *Daphnia* [62], and in the oyster *Saccostrea glomerate* [63], where a decrease in the amount of energy allocated to eggs due to direct exposure to contaminant led to smaller offspring. Therefore, we might describe here a cascading model of changes in the life-history traits of *G. fossarum* that originates from a reduction of F1 brood size, which then has resulted in an increased energy allocation to reproduction in F2 females, and then a reduced growth in F3 juveniles.

### 2.4. Mitigation of the Demographic Impact along Generations

We developed a population model to assess the demographic consequences of the modifications of life-history traits recorded across the generations. We followed a modeling approach adapted from Coulaud et al. [34] who developed a population dynamics model for a field population of *G. fossarum*. Here, the model is calibrated to reflect the demographics of a laboratory population and does not take into account, for example, the effects of interspecies relationships (competition, predation) that occur under natural conditions. As in the study by Coulaud et al. [34], one of the perspectives could be to deepen the approach to take into account ecological elements that are closer to the constraints exerted on the field populations (e.g., seasonal phenology). In the present study, the matrix model is used not as a tool for forecasting population dynamics but as a projection tool at the population level [64], allowing an integrated analysis of the potential demographic weight of the changes observed in life-history traits. The analysis of the models calibrated for the generations F1, F2, and F3 of the control condition provided weekly asymptotic population growth rates of 1.09, 1.07, and 1.07, respectively (Figure 5). In addition, for each generation, the population stable structure in control presented a pyramid shape, with most individuals in juvenile classes (Figure 5). This is consistent with the size distribution we observed in a field population of gammarids [34]. These results indicate that the rearing conditions in the laboratory allowed properly maintaining the control populations in a steady state over the one-year experiment period. The changes in life-history traits of the generation F1 resulted in an 11% and 6% decrease in the weekly asymptotic population growth rate compared to control for the 0.3 µg Cd/L and 3 µg Cd/L conditions, respectively (Figure 5). Parental exposure also modified the stable structure in F1 (Figure 5). By contrast, the asymptotic population growth rates calculated for the generations F2 and F3 of the condition 3Cd-F0 were close to the control conditions. The F2 and F3 population stable structures of 3Cd-F0 were also similar to the structure of the control population with the restoration of high percentages of individuals in juvenile classes compared to the F1 structure (Figure 5). Hence, the restored fertility in F2 compared to F1 females, along with the early puberty in F2 individuals and the decrease in size at puberty in F2 offspring, supported a mitigation trend of demographic impact along generations. The reduced growth ability of F3 organisms did not seem to affect the population dynamics, even if this result has to be regarded with caution, since the parametrization of F3 fertility rates was based on F2 data.

Such a mitigation of adverse demographic effects was previously suggested in other invertebrates, e.g., in *Paronychiurus kimi* populations exposed to paraquat [10], or in *Plectus acuminatus* populations exposed to pentachlorophenol [65]. In the same vein, Prud’homme et al. [11] showed in the mosquito *Aedes aegypti* that changes in offspring life-history traits induced by ancestral exposure to ibuprofen or benzo[a]pyrene led to negligible population consequences due to a compensation between increased mortality, accelerated development, and female-biased sex ratio within offspring generation. In our study, compensatory processes involving trade-offs between life-history traits of successive generations (e.g., a reduced F1 fertility against a greater puberty investment in F2) allowed finally mitigating the demographic impact of Cd exposure in the offspring generations. Moreover, even if these compensatory processes triggered a mitigation of the demographic impact of F0 exposure, they have also profoundly modified the life history of the population, particularly with a severe reduction of F3 organism sizes. Interestingly, we previously identified a field *G. fossarum* population historically exposed to Cd that presents a reduction in the average body size of adult organisms [28]. Transgenerational processes were also shown to be involved in the development of Cd tolerance in this population [7]. Such changes in the distribution of population body size can have significant consequences on the functioning of the aquatic ecosystem—for example, for the recycling of organic matter by affecting the shredding activity of detritus by macroinvertebrates [66,67] or for the availability of biomass for aquatic predators [68].

## 3. Materials and Methods

### 3.1. Test Protocol

*Collection of F0 spawners:* Gammarids were harvested in a wild population living in an upstream part of the Bourbre River (Isère, France) in March 2017 by using a hand-held net. They were sieved (2–2.5 mm) in the field to target adult size classes in the sampling. We quickly brought them back to the laboratory. We maintained them for 24 h before the start of the experiment at 12 °C under a 16/8 h light/dark cycle into 20 L aquaria (4000 organisms/aquarium) continuously supplied thanks to a flow-through-system with groundwater continuously pumped, filtered, and aerated in our laboratory facilities (mineral composition: [HCO_3_^−^] = 220 mg/L; [Cl^−^] = 12 mg/L; [SO_4_^2−^] = 37 mg/L; [Na^+^] = 9 mg/L; [K^+^] = 2 mg/L; [Mg^2+^] = 7.5 mg/L; [Ca^2+^] = 75 mg/L; [NH_4_^+^] < 0.02 mg/L; [NO_2_^−^] < 0.05 mg/L; [NO_3_^−^] = 7 mg/L; pH = 8; conductivity = 450 µS/cm). Then, 1200 precopulatory pairs with females in the D2 molt stage were selected and were equally assigned to one of the three exposure conditions (i.e., control; 0.3 µg Cd/L; 3 µg Cd/L).

*F0 exposure to Cd*: For each of three exposure conditions, 400 precopulatory pairs were then placed in a 12 L bucket with continuously oxygenated test solution (Appendix A). F0 exposure was carried out over 21 days, i.e., during one gametogenesis cycle (Figure 1). The test solutions (control; 0.3 µg Cd/L; 3 µg Cd/L) were prepared from two Cd stock solutions (CdCl_2_(2^1/2^H_2_O) Sigma Aldrich^®®^), respectively 1.5 mg Cd/L and 15 mg Cd/L in milliQ water and renewed twice a day. For each solution renewal, 2 mL of stock solutions were added to 10 L of uncontaminated drilling water previously oxygenated during 24 h at 16 °C.

*Offspring breeding:* At the end of parental exposure, F0 gammarids were placed in three buckets of 12 L (one bucket per condition) continuously supplied with uncontaminated drilling water (4 renewals per day). We discarded the minority of females that were delayed in their reproductive cycle or without embryos. One week before the estimated date for F1 juvenile release, F0 females were placed in 2 L tanks, and F0 males were discarded to avoid predation on neonates. The tanks were supplied continuously with uncontaminated drilling water (50 renewals per day). Released F1 juveniles were collected once a week during two successive weeks. Hence, the deviation of the age between F1 juveniles collected in a given condition did not exceed 15 days. Five months after F1 release, F2 juveniles were collected, and F3 juveniles were collected 6 months later following the same protocol. For each generation, the collected juveniles were counted on a light table.

During the whole experiment, water temperature was set at 16 °C. Adult gammarids and their offspring were fed *ad libitum* on alder leaves with the addition of gammarid feces for neonates. They also received dried tubifex worms as a complementary food resource. Pieces of plastic mesh were placed in each container as shelters.

### 3.2. Molecular and Physiological Measurements

*Genome integrity:* Sperm DNA damages were assessed through the alkaline version of comet assay by adapting the procedure described in *G. fossarum* by Lacaze et al. (2010). After testis dissection of mature male gammarids (*n* = 10 per condition), 80 µL of 0.625% low-melting agarose in PBS (37 °C) were mixed with 20 µL of cell suspension collected from one organism. Two 40 µL drops of this mixture were deposited onto an agarose gel-coated slide enabling further DNA damage measurement on 2 × 50 cells per testis sample.

*Epigenetic mark:* Global cytosine methylation levels in entire organisms were measured in males following the procedure described in Cribiu et al. [29].

*Feeding assay*: The feeding rate of F0 organisms during exposure was assessed according to the procedure described in Coulaud et al. [69]. Twenty male gammarids and 20 leaf discs (20 mm diameter) were enclosed in perforated cylinders (four replicates per condition), which were then placed in the buckets during Cd exposure. After 7 days, leaf discs were collected and numerically scanned to estimate leaf consumption. The procedure was slightly modified for the F1 generation (5 months old) to preserve the stock of spawners dedicated to pursuing the experiment. Three replicates of 4 males/4 leaf discs were considered per condition. The feeding rate was not monitored for the F2 and F3 generations.

*Growth:* Gammarid juveniles were photographed under a binocular light microscope at a three-week interval. Then, photographs were analyzed using the SigmaScan® Pro v5.0 software (Systat Software, San Jose, CA, USA). The body size corresponds to the dorsal length from the start of the prosoma to the end of the metasoma.

*Puberty monitoring:* Eight weeks after the first juvenile collection of each generation, the organisms were observed once a week on the light table in order to identify female individuals reaching puberty (presence of newly produced oocytes in ovaries). Then, the mature females were placed in another tank with a similar number of males (larger in size). The percentage of puberty was calculated by dividing the cumulative number of mature females by the total number of mature females collected at the end of puberty monitoring.

*Reproductive capacity*: The molt stage, the number of oocytes, and the number of embryos per female as well as the secondary follicle surface (used as a proxy of yolk reserves) were evaluated as described by Geffard et al. [35]. The number of embryos of D2 females was not taken into account (release of neonates at this stage). For the number of oocytes, corrective factors established from Geffard et al. [35] were applied to consider the influence of molt stage (e.g., 30% reduction between C1 and C2 stages). Since the number of oocytes and embryos depends on female body size, standardization has been performed as proposed in Geffard et al. [35] by dividing the total count of oocytes by female body size and those of embryos by female body size minus size at puberty (5 mm). Regarding the analysis of the secondary follicle surface, females without oocytes were discarded from the dataset, and females at B and C1 molt stages were not considered, since secondary vitellogenesis is not initiated in these stages. Corrective factors from the study of Geffard et al. [35] were applied to make it possible to compare follicle surface measurements between females at the C2, D1, and D2 molt stages (using C2 as reference). The sperm viability of male gammarids was studied using the methodology of Lacaze et al. [37].

*Timing of biological measurements along generations* (Appendix A)*:* In order to assess the direct effects of parental exposure, (1) a feeding assay was performed with F0 males 11 days after the beginning of Cd exposure; (2) on the 18^th^ day of exposure, the females were randomly selected for measuring reproductive markers, i.e., the molt stage, the number of oocytes (F1 in germ cell stage), the number of embryos and the secondary follicle surface (used as a proxy of yolk reserves); (3) 20 days after the beginning of exposure, 30 F0 males from each condition were also sampled at the precopulatory stage (before the fertilization of exposed gametes), and 10 of them were used for sperm viability assessment, 10 were used for sperm genotoxicity measurement (comet-assay), and 10 were used for assessing global DNA methylation levels. In F1 offspring, the survival and growth of juveniles were monitored. A feeding assay was conducted after puberty, and several mature organisms were sacrificed (after collecting F2 neonates) to measure sperm viability, DNA damage, global DNA methylation levels, and reproductive endpoints in females. As in F1 offspring, the survival and growth of F2 and F3 juveniles were monitored. The puberty progress was evaluated in F2 organisms, but it was not possible to assess sperm viability, global DNA methylation levels, and feeding rates in males of F2 and F3 generations for organizational reasons. As a result of the long-term study, the reproductive cycles in F2 generations were desynchronized between females. Hence, contrary to the random selection at one date for F0 and F1 females, F2 females were monitored three times a week for measuring reproductive endpoints in C2–D1 females. Then, the mature males in precopulatory pairs with the sacrificed females were placed in tanks with mature females from a lab breeding stock for stimulating the spermatogenesis until the genotoxicity assessment (comet assay).

*Statistical analyses:* We carried out statistical procedures with the R software (version 3.5.1). For quantitative data (DNA damage, global methylation levels, feeding rate, growth rate of juveniles, body size at puberty, and reproductive capacity endpoints), Kruskal–Wallis tests (KW) were used followed by Mann–Whitney tests (MW) for paired comparisons. Offspring survival was compared by applying proportion tests. The significance level used for all statistical tests was 0.05.

### 3.3. Population Dynamics Modeling

A reference life cycle graph of *G. fossarum* (Appendix A) was defined to establish a size structured Lefkovitch matrix model [64]. The matrix model was parametrized for each condition and each generation. Only females were taken into account, and the population was structured in 7 size classes: embryos (*class 1*), two juvenile classes (*class 2*, individuals with a size up to 2.5 mm and *class 3*, size from 3.5 to 5 mm), and four adult classes (class *4* size from 5 to 6.5 mm, *class 5* size from 6.5 to 8 mm, *class 6* size from 8 to 9.5 mm, and *class 7* greater than 9,5 mm) (Appendix A). The parameterization was calibrated on a time step of one week and a post-breeding census hypothesis. Based on the life cycle graph of *G. fossarum* (Appendix A), the following projection matrix **A** was established:(1)$$A=(\begin{array}{ccccccc} P1 & 0 & 0 & F4 & F5 & F6 & F7\\ G1 & P2 & 0 & 0 & 0 & 0 & 0\\ 0 & G2 & P3 & 0 & 0 & 0 & 0\\ 0 & 0 & G3 & P4 & 0 & 0 & 0\\ 0 & 0 & 0 & G4 & P5 & 0 & 0\\ 0 & 0 & 0 & 0 & G5 & P6 & 0\\ 0 & 0 & 0 & 0 & 0 & G6 & P7 \end{array})$$
where *P_i_* is the proportion of surviving organisms remaining in size class *i*, *G_i_* is the proportion of surviving individuals moving from size class *i* to size class *i* + 1, and *F_i_* is the fertility, i.e., the number of female oocytes produced per adult female during a time step. The first dominant eigenvalue of the matrix **A** corresponds to the asymptotic population growth rate λ, and the right eigenvector **w** was associated with λ to the stable structure of the population. *P_i_* and *G_i_* parameters were expressed as described by Caswell (2001) as follows:(2)$$G_{i}=\sigma_{i}\gamma_{i}$$
(3)$$P_{i}=\sigma_{i}\;\left(1-\gamma_{i}\right)$$
where $\sigma_{i}$ is the probability of survival after a time step in size class *i*, and $\gamma_{i}$ is the transition rate, i.e., the probability that individuals grow from size class *i* to size class *i*+1 after one week. The transition rates $\gamma_{i}$, which depend on λ, were calculated by applying the iterative method proposed by Caswell [64] according to the following formula:(4)$$\gamma_{i}=\frac{{\left(\frac{\sigma_{i}}{\lambda }\right)}^{d_{i}-1}}{1+\left(\frac{\sigma_{i}}{\lambda }\right)+{\left(\frac{\sigma_{i}}{\lambda }\right)}^{2}+\cdots +{\left(\frac{\sigma_{i}}{\lambda }\right)}^{d_{i}-1}\;}$$
where *d_i_* is the duration of size class *i*. We assumed that the fertilization was not limited by male availability and sperm viability in the different conditions. As a result of the post-breeding census, fertility entries $F_{i}$ of the matrix **A** were then calculated for the adult classes (*i =* 4, 5, 6, 7) from $\sigma_{i}$, the sex ratio (*sr*), the proportion of reproductive females (*rf_i_*), the average number of oocytes per female (*no_i_*), and the duration of the female reproductive cycle (*d_c_*):(5)$$F_{i}=\sigma_{i}\;\left(sr\times rf_{i}\times no_{i}\right)/d_{c}.$$

For each projection matrix (each generation in each condition), $\sigma_{i}$ (for *i* > 1) values were calculated from survival curves established during the experiment, and $\sigma_{1}$ (proportion of surviving embryos) corresponds to the ratio between the number of embryos and oocytes. The duration of the size class *i* (*d_i_*) was calculated based on growth curves. Regarding the calculation of the fertility *F_i_*, the sex ratio was set at 0.5. Given the experimental temperature of 16 °C, the duration of the reproductive cycle (*d_c_*) was set at 3 weeks [36]. The proportion of reproductive females was determined from puberty monitoring data. The average number of oocytes per female of each size class was estimated from reproductive data recorded in the different conditions for F1 and F2 generations. An extrapolation of recorded measurements to the different size classes was operated, based on the mean size of each class, and a linear relationship was established between the oocyte number and body size of females. This relationship was obtained from previous fertility data recorded on the same source population in different studies from our laboratory:*no* = 2.23 × body size.(6)

Since the experiment was stopped before the F3 offspring reached puberty due to technical problems in our water supply facilities, the proportion of reproductive females and oocyte number in the F3 generation were not recorded. They were estimated from the F2 control records. We conducted all calculations and algebraic computations with R software (version 3.5.1) (R Foundation for Statistical Computing, Vienna, Austria).

## 4. Conclusions

Despite the shortness of F0 exposure to environmental Cd concentrations and the absence of visible F1 impairments until the adult stage, significant cascading effects on the life history of *G. fossarum* occurred along the three generations. These first findings need further confirmation in future experimental replications. Nevertheless, they highlight the need for considering successive generations and environmentally relevant species with diversified life-history strategies in ecotoxicological testing in order to improve the predictive approaches in ecological risk assessment. Furthermore, the contrast between the weak demographic alterations in offspring generations and the severe modifications of the life-history traits with potential consequences for the ecosystem (e.g., trophic chain imbalance), leads us to point out the role of population compensatory processes that might shape the long-term responses of populations to environmental toxicity.

## Figures and Tables

**Figure 1 ijms-21-04704-f001:**
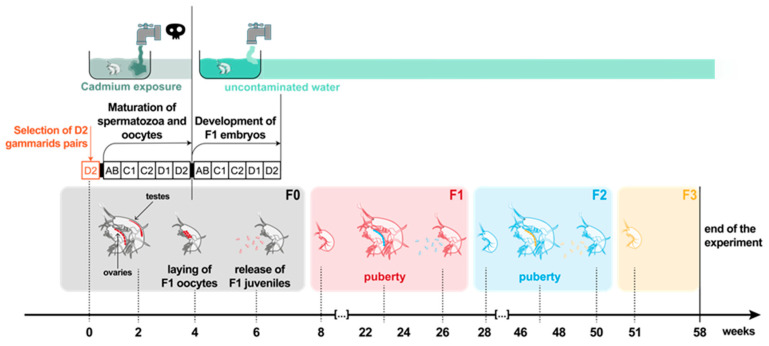
General scheme of the transgenerational study. The exposure of F0 genitors to Cd was maintained during one gametogenesis cycle. Each generation is shown with a specific color (red for F1, blue for F2, yellow for F3) from germline cells up to the adult stage. Small, medium, and large body size organisms represent immature gammarids, reproductive females, and males, respectively.

**Figure 2 ijms-21-04704-f002:**
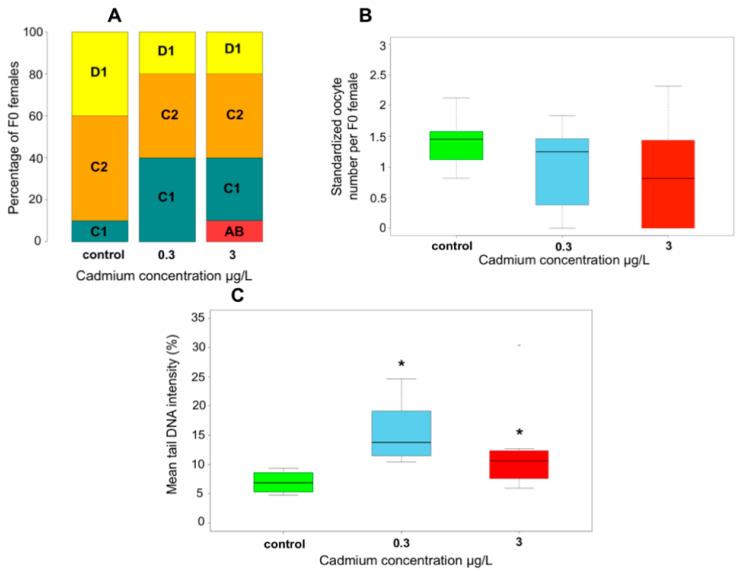
F0 responses to Cd exposure. (**A**) Molt stage distribution of F0 females at the end of exposure, *n* = 10. (**B**) Oocyte production in F0 females (size-standardized oocyte number per female, *n* = 10). (**C**) Sperm DNA damages of F0 males (comet assay, mean tail DNA intensity), control: *n* = 9, 0.3 µg Cd/L: *n* = 8, 3 µg Cd/L: *n* = 8. Green, blue, and red colors in boxplots (B) and (C) correspond to control, 0.3 µg Cd/L, and 3 µg Cd/L, respectively; the boxes extend from the first to the third quartile of data distribution with a bold segment for the median value; the whiskers extend to the extreme data points (excluding outliers distant over 1.5 times the interquartile range from the median); star denotes a significant difference compared to the control condition (*p*-value < 0.05).

**Figure 3 ijms-21-04704-f003:**
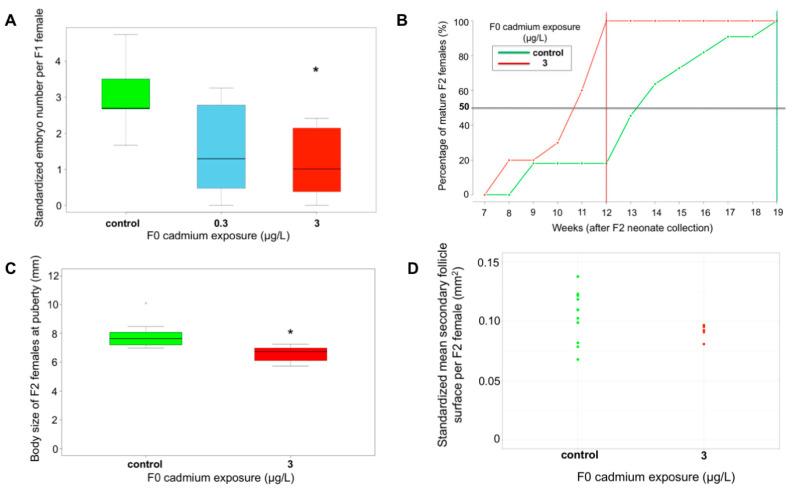
Effects of F0 exposure on F1 and F2 reproductive capacity. (**A**) Embryo number per female in F1 generation (size-standardized embryo number), control: *n* = 9, 0.3 µg Cd/L: *n* = 7, 3 µg Cd/L: *n* = 8. (**B**) Dynamics of puberty during F2 generation (cumulative number of mature females/total female number), control: *n* = 11, 3 µg Cd/L: *n* = 10. (**C**) Body size of F2 female at puberty, control: *n* = 11, 3 µg Cd/L: *n* = 10. (**D**) Mean secondary follicle surface per F2 female, control: *n* = 11, 3 µg Cd/L: *n* = 7. Star denotes a significant difference compared to the control condition, *p*-value < 0.05.

**Figure 4 ijms-21-04704-f004:**
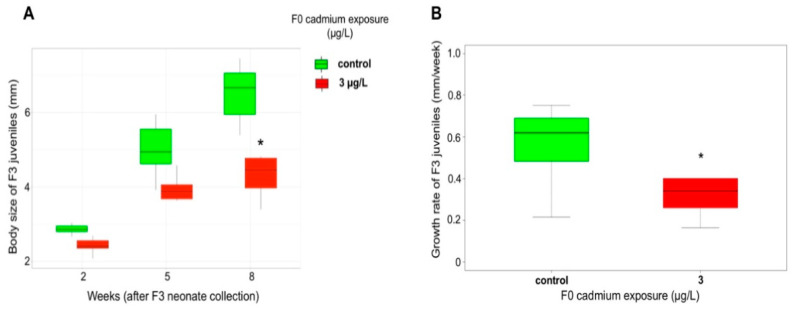
Effects of F0 exposure on F3 juveniles. (**A**) Body size of F3 juveniles, *n* = 10. (**B**) Weekly growth rate of F3 juveniles, *n* = 10. Star denotes a significant difference compared to the control condition, *p*-value < 0.05.

**Figure 5 ijms-21-04704-f005:**
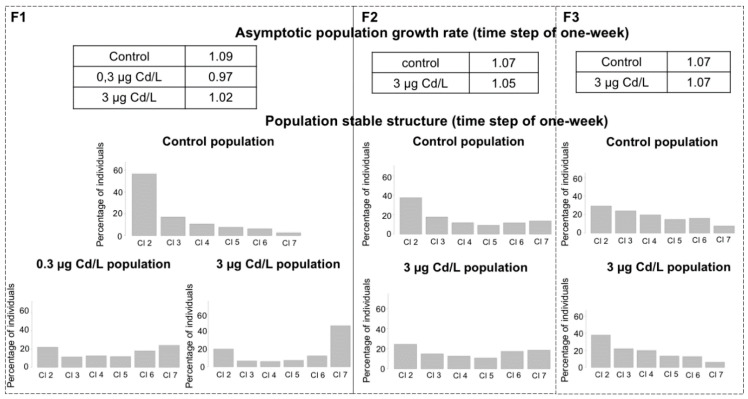
Projection of population effects of F0 exposure. Demographic indicators (the asymptotic population growth rate and the population stable structure) were calculated for F1, F2, and F3 generations (time step of one-week). Cl = age class. Appendix A presents the life cycle graph corresponding to the Lefkovitch model used for the calculation.

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
