# Peer review of "A “Population Dynamics” Perspective on the Delayed Life-History Effects of Environmental Contaminations: An Illustration with a Preliminary Study of Cadmium Transgenerational Effects over Three Generations in the Crustacean Gammarus"

_ijms, 2020, doi:10.3390/ijms21134704_

Round 1

Reviewer 1 Report

This manuscript was submitted to the IJMS as a contribution to the special issue "Advances in Metal Metabolism Research” edited by Prof Reinhard Dallinger (myself).

The authors applied a multi-generation exposure experiment with Cd2+ to the freshwater gammarid Gammarus fossarum, a species that has attained during the last years an increasing significance, among others, in biomonitoring of environmental pollution and ecotoxicology. In the experiment, the influence of Cd at two concentrations (0.3 and 3 µg/L) is traced from a Cd-exposed parental gammarid generation (F0) through three unexposed succeeding generations (F1, F2 and F3). Cd effects through four generations of Gammarus fossarum were assessed by  including genotoxic (Comet assay for detecting mean tail DNA integrity), epigenetic (DNA methylation), reproductive (broodsize, molt stage, secondary follicle surface, sperm viability), physiological (survival, growth and growth rates) and population structure analyses (applying a Lefkovitch matrix model). The main results show that, although significant effects on some of the assessed parameters were detected through generations, the population structure model indicated that the transgenerational effects of Cd had only weak impacts on a demographic level.

Overall, the study is robust and the results are sound and interesting, although I’m aware of the fact that the present manuscript does not directly address questions of metal metabolism. In spite of this, I would like to include this paper in the special issue.

I have a few questions to the authors regarding their experimental concept and discussion of their results.

On page 5 (lines 173-175), the authors report that the survival of F1 juveniles was significantly lower than in controls at an exposure level of 0.3 µg Cd / L, but not at the higher exposure level of 3 µg Cd / L. The authors argue that this might be due to the high variability of this parameter in natural populations. I agree. At this point, however, one has to remind that the experimental setup was based on only one experiment, without biological replicates. The authors might mention this in their discussion.

On pages 6 and 7 (lines 205-236), the authors discuss the observed effects in their Cd experiments, and compare them with supposedly similar effects of other contaminants (atrazine, dioxine, etc.) in other species. This approach is eligible. I would expect, however, that the authors might defer more accurately on the specific metabolic effects of Cd, since similar effects of very different compounds at higher levels of biological organization (e.g. reproduction) will certainly mask more compound-specific effects at the metabolic levels. I stongly suggest that the authors discuss more extensively the expected and/or known specific metabolic effects of Cd on the life history parameters of Gammarus fossarum, in particular, such as the potential effect of Cd on endocrine balance. One could also ask, moreover, if specific effects of Cd may have different impacts on energy allocation strategies in adult and juvenile Gammarus fossarum.

On page 7-9 (lines 264-307), the authors discuss the mitigation of the demographic impact along generations. This is a very strong approach. However, can this demographic model also be applied to populations in the field? I wonder whether retardation and reduction of growth in juveniles (as shown for F2 females) could lead to an alteration in feeding patterns of predators that feed on Gammarus fossarum? I’m asking this because we know that in some food chains, alterations in individual growth and size can expose some species to a higher risk of predation, and the consequences of such alterations on population dynamics are still open. Does the present model take this into account? If not, the authors might discuss this argument in their manuscript.

On page 4, Figure 2 needs some specifications. I suggest that the authors may add to the figure legend explanations of colors and abbreviations in the bars of Figure 2A.

The authors may also check their manuscript for typing errors before re-submission of an improved version.

Author Response

- Reviewer 1

Comments to the Author

The authors applied a multi-generation exposure experiment with Cd2+ to the freshwater gammarid Gammarus fossarum, a species that has attained during the last years an increasing significance, among others, in biomonitoring of environmental pollution and ecotoxicology. In the experiment, the influence of Cd at two concentrations (0.3 and 3 µg/L) is traced from a Cd-exposed parental gammarid generation (F0) through three unexposed succeeding generations (F1, F2 and F3). Cd effects through four generations of Gammarus fossarum were assessed by  including genotoxic (Comet assay for detecting mean tail DNA integrity), epigenetic (DNA methylation), reproductive (broodsize, molt stage, secondary follicle surface, sperm viability), physiological (survival, growth and growth rates) and population structure analyses (applying a Lefkovitch matrix model). The main results show that, although significant effects on some of the assessed parameters were detected through generations, the population structure model indicated that the transgenerational effects of Cd had only weak impacts on a demographic level.

Overall, the study is robust and the results are sound and interesting, although I’m aware of the fact that the present manuscript does not directly address questions of metal metabolism. In spite of this, I would like to include this paper in the special issue.

I have a few questions to the authors regarding their experimental concept and discussion of their results.

On page 5 (lines 173-175), the authors report that the survival of F1 juveniles was significantly lower than in controls at an exposure level of 0.3 µg Cd / L, but not at the higher exposure level of 3 µg Cd / L. The authors argue that this might be due to the high variability of this parameter in natural populations. I agree. At this point, however, one has to remind that the experimental setup was based on only one experiment, without biological replicates. The authors might mention this in their discussion.

-> We definitely agree with this comment and we now mention this point (L181-183), which was already highlighted in a sense in the first paragraph of results (L126-128).

On pages 6 and 7 (lines 205-236), the authors discuss the observed effects in their Cd experiments, and compare them with supposedly similar effects of other contaminants (atrazine, dioxine, etc.) in other species. This approach is eligible. I would expect, however, that the authors might defer more accurately on the specific metabolic effects of Cd, since similar effects of very different compounds at higher levels of biological organization (e.g. reproduction) will certainly mask more compound-specific effects at the metabolic levels. I stongly suggest that the authors discuss more extensively the expected and/or known specific metabolic effects of Cd on the life history parameters of Gammarus fossarum, in particular, such as the potential effect of Cd on endocrine balance. One could also ask, moreover, if specific effects of Cd may have different impacts on energy allocation strategies in adult and juvenile Gammarus fossarum.

-> We understand this request and we have now discussed this point in more details regarding previous results obtained on the potential ED mode of action of Cd in Gammarus which could support our present results on F1 (L216-233; L254-261).

On page 7-9 (lines 264-307), the authors discuss the mitigation of the demographic impact along generations. This is a very strong approach. However, can this demographic model also be applied to populations in the field? I wonder whether retardation and reduction of growth in juveniles (as shown for F2 females) could lead to an alteration in feeding patterns of predators that feed on Gammarus fossarum? I’m asking this because we know that in some food chains, alterations in individual growth and size can expose some species to a higher risk of predation, and the consequences of such alterations on population dynamics are still open. Does the present model take this into account? If not, the authors might discuss this argument in their manuscript.

-> Indeed, the theoretical model used here applies to a demography in a laboratory context. Interspecific interactions are clearly not taken into account at this stage of the analysis of demographic effects. We have now better highlighted this point in this revised version (L293-300). On the other hand, it is exactly this type of argument that we thought it important to stress in our first version (in discussion L336-339), where we mentioned that the modification of life history traits (smaller growth ability in particular), even if it results in demographic compensation at the population level, can generate strong indirect effects particularly on the trophic balance within the ecosystem (we reinforced this point in the conclusion L486-489).

On page 4, Figure 2 needs some specifications. I suggest that the authors may add to the figure legend explanations of colors and abbreviations in the bars of Figure 2A.

-> We added these specifications in the legend of Figure 2.

The authors may also check their manuscript for typing errors before re-submission of an improved version.

-> We carefully checked the manuscript for typing errors.

Reviewer 2 Report

Dear authors,

congratulations to your paper. Your results are exciting and it is very interesting to see long-term effects of Cd exposure in gammarids. It's a fantastic work!

This study is well conducted and clearly presented. All methods and results are very well discussed and I don’t have any major concerns. In my opinion, the paper can be accepted after some minor revisions.

L 315: could you please add some more information about the "drilling water"? Is this artificial water, tap water, filtered water….? Do you have some some information about the composition of minerals as well as conductibility and pH? Did you maintain the gammarids in flow-through-systems? 

Minor typo errors:

L 30: add comma before and after “however”

L 144: add comma after “Nevertheless”

L 218: add blank before “0.31”

L 245: delete blank between “5” and “%”

L 248: “number of embryos” instead of “number o embryos”

Thank you!

Author Response

Reviewer 2:

Dear authors,

congratulations to your paper. Your results are exciting and it is very interesting to see long-term effects of Cd exposure in gammarids. It's a fantastic work!

This study is well conducted and clearly presented. All methods and results are very well discussed and I don’t have any major concerns. In my opinion, the paper can be accepted after some minor revisions.

-> We thank the reviewer for his enthusiastic comment.

L 315: could you please add some more information about the "drilling water"? Is this artificial water, tap water, filtered water….? Do you have some some information about the composition of minerals as well as conductibility and pH? Did you maintain the gammarids in flow-through-systems? 

-> We have added these technical specifications in the M&M (L346-351).

Minor typo errors:

L 30: add comma before and after “however”

L 144: add comma after “Nevertheless”

L 218: add blank before “0.31”

L 245: delete blank between “5” and “%”

L 248: “number of embryos” instead of “number o embryos”

-> We brought all these typo corrections to the MS.